# Design Optimization of a HAZMAT Multimodal Hub-and-Spoke Network with Detour

**DOI:** 10.3390/ijerph182312470

**Published:** 2021-11-26

**Authors:** Shuxia Li, Yuedan Zu, Huimin Fang, Liping Liu, Tijun Fan

**Affiliations:** School of Business, East China University of Science and Technology, Shanghai 200237, China; sxli@ecust.edu.cn (S.L.); zyd11250306@163.com (Y.Z.); Y30181309@mail.ecust.edu.cn (H.F.); lpliu@ecust.edu.cn (L.L.)

**Keywords:** hazardous materials, multimodal transport, hub-and-spoke network, detour strategy, risk quantification

## Abstract

The growing transportation risk of hazardous materials (hazmat) is an important threat to public safety. As an efficient and reliable mode of transportation, the multimodal hub-and-spoke transport network helps to achieve economies of scale and reduce costs. Considering the dual goals of risk and cost management of hazmat transportation, a novel optimization model of a multimodal hub-and-spoke network with detour (MHSNWD) for hazmat on the strategic level is designed. It integrates the planning of hub location and route selection based on the risk quantification for different transportation modes. Additionally, a detour strategy is applied, which allows for more than two hub nodes to be selected to form an optimal path between any supply and demand nodes in a hub-and-spoke network. Then, the risk is taken as the main objective and the cost is converted into a budget constraint to solve the model by using CPLEX. Additionally, a numerical study is conducted based on a CAB dataset to find the influence of the number of hubs and budget constraints on the optimization results. In addition, a counterpart model of the multimodal hub-and-spoke network without detour (MHSNOD) is tested to validate the advantages of the proposed model of MHSNWD. The numerical experiment shows that an appropriate increase in the number of hubs and the cost budget can remarkably reduce network risk. Compared with MHSNOD, the optimal result of MHSNWD can achieve a marginal improvement in risk reduction. This work may provide an informative decision-making reference for planning a hazmat transportation network.

## 1. Introduction

As raw materials for industrial production, hazardous materials (hazmat) play an important role in modern industrialized society. With the rapid development of the petrochemical industry and continuous extension of its industrial chain, the hazmat industry has derived a large number of logistics requirements. According to statistics, the market size of hazmat logistics in China exceeded CNY 2 trillion in 2020 and the transport volume of hazmat every year in China shows an increasing trend [1]. The growing transport demand means that the hazmat logistics industry faces great challenges. Moreover, due to the physical and chemical characteristics of hazmat, such as its inflammable, explosive, corrosive, and highly toxic properties, mobile hazardous sources in the process of transportation will come with unpredictable and irregular social and environmental risks, posing a serious threat to people’s lives and property safety [2,3,4]. The risk of hazmat transportation incidents has also increased worldwide. According to China’s Ministry of Emergency Management, from 2010 to 2015, more than 400 major accidents occurred in China, about 80% of which were caused by transportation and 80% of which involved hazmat. In 2019, there were also over 22,000 hazmat incidents by different transportation modes in the US [5]. Therefore, hazmat carriers need to optimize the transport network to reduce the risks as much as possible.

In reality, point-to-point direct transport has brought many problems, such as high operations costs [6]. Consequently, modern logistics is rapidly developing towards complex networks to meet the increasing transportation demand. Additionally, the hub-and-spoke network is one of the most widely used networks in many industries. To achieve economies of scale, appropriate nodes are usually selected as hubs for the collection, consolidation, transfer, and distribution of goods. In addition, hazmat is often stored at production sites and their final destinations are far away in most cases, which makes the need to travel long distances inevitable. For example, approximately 95% of hazardous materials in China are moved via long-distance transportation [7]. Since rail services are safer and cheaper than road services, multimodal transport as an efficient, reliable, and sustainable method has become the mainstream practice for the long-distance and large-scale transportation of hazmat internationally [8]. The European Commission launched the Marco Polo project in 2014 to facilitate the transition from single-road transportation to multimodal transportation. It is estimated that 30% and 50% of the total European transport volume will be covered by multimodal transportation in 2030 and 2050, respectively [8]. In 2016, China’s Ministry of Transport also issued several guidelines on promoting the development of multimodal transportation for an international logistics “corridor” with improved transportation quality and efficiency. As a result, the hazmat multimodal hub-and-spoke network design problem has attracted the interest of industries, governments, and academia.

The traditional hub network problems for regular freight transportation are mostly aimed at minimizing the total transportation cost [9,10,11,12]. Considering transportation cost is proportional to the distance travelled, it is generally assumed that arc lengths between hub nodes satisfy the triangle inequality [13,14], which means that direct hub connections are more economical than detours. Thus, no more than two hub nodes will appear in an optimal path between any pair of supply and demand nodes in a hub-and-spoke network. However, both hazmat multimodal companies and the government are under great pressure to consider transport-related risk [15,16,17,18,19]. Since a direct link between two hubs may pass through a densely populated area, causing a higher risk, a detour route is considered in this paper to form a less risky network. Meanwhile, two commonly used transport modes, i.e., road and railway, are taken into account to realize the objectives of cost and risk management in the hazmat multimodal hub-and spoke network.

The rest of this paper is organized as follows. Section 2 provides a comprehensive literature review on the hazmat multimodal network design and research gaps. Section 3 presents a design problem of a hazmat multimodal hub-and-spoke network with detour and the corresponding models under two strategies are constructed. The numerical study based on a CAB dataset is given in Section 4. Finally, conclusions are drawn in Section 5.

## 2. Literature Review

With the sustainable development of the environment, economy, and society, sustainable freight transportation has become an important topic that needs to be thoroughly considered [20]. In addition, transportation companies also wish to avoid economic and reputational damage and risks [21,22,23]. These factors push for the traditional transportation mode to be changed with the requirements of efficiency and risk management [24], especially in the hazmat transport industry [25,26,27].

Using a single direct mode of transport makes it hard to meet the transportation needs of large volumes of goods and the characteristics of the continuous production of hazmat mean that the production sites must be located far from customers. Therefore, multi-modal transportation as an efficient long-distance transportation mode is experiencing a period of great development potential, which has aroused the interest of many scholars for its utility in solving the hazmat transport problem [28]. Verma et al. [15,29] claimed that there is an unignorable gap between hazmat intermodal transportation practice and academic activity. Additionally, they presented a first attempt at the development of an analytical framework for planning the rail–truck intermodal transportation of hazmat. Xie et al. [30] proposed an innovative multimodal hazmat location and routing model for a realistically sized interconnected network to optimize transfer yard locations and routing plans simultaneously. In addition, Assadipour et al. [31] presented a bi-objective hazmat optimization framework for managing hazmat freight that not only considers congestion at intermodal yards but also determines the appropriate equipment capacity to further reduce the congestion and network risk. Sun et al. [32] explored the hazmat multimodal routing problem from the operational level of network planning and formulated a comprehensive model concerning characteristics, such as a capacitated schedule-based rail service and an uncapacitated time-flexible road service, environmental risk constraints, etc. Ghaderi and Burdett [33] developed a generic strategic planning approach for transporting hazmat in a bi-modal transportation network to reduce the costs and risks. The location of the transfer facilities, which are subject to probabilistic disruption, and the integrated routing problem are formulated as a two-stage stochastic programming model and solved by three different heuristics based on sampling. Fontaine et al. [34] investigated a bi-level multimode hazmat transport network design problem by considering population-based risk equilibration, where the government wants to equilibrate the risk by restricting links to the shipment of hazmat and anticipate the reaction of the carriers who want to minimize the transportation costs.

For the hazmat multimodal transport network design problem, as described above, the underlying structure of the network is a straight-through network. However, there are also many other studies concerning hub networks. Since the seminal paper by O’Kelly [35], a considerable amount of work about hub network designs (see, e.g., [36,37,38,39]) has been conducted in the past 30 years. Additionally, many scholars began to combine the hub location problem with multimodal transportation. Ishfaq and Sox [40] pointed out that the design of intermodal transportation networks becomes more complex than for single-mode logistics. Additionally, the effects and interactions of several factors, such as the transportation cost structure, modal connectivity, availability of hubs, and service time performance, on the design of intermodal hub networks are studied in depth. Alumur [41] introduced a multimodal hub location problem from a network design perspective and considered both transportation costs and travel times to meet service time promises between origin and destination pairs. Additionally, the study proposed adding the pricing issues and market competition in future works. Ambrosino et al. [12] further incorporated capacity bounds both for the candidate hub nodes and the arcs incident to them into the hub location problem arising in a freight logistics multimodal network.

Moreover, a hub-and-spoke network has also been adopted in multimodal transport due to its advantages in profit maximization [42,43,44,45]. Yang et al. [46] developed a novel modeling framework for the planning and optimization of an intermodal hub-and-spoke network considering mixed uncertainties in both transportation cost and travel time. Shang et al. [47] constructed a hierarchical multimodal hub-and-spoke distribution network for cargo delivery systems with uncertainties in travel time and handling time, which aims to minimize the latest arrival time under the diverse credibility of chance constraints. Tang et al. [6] established a dual-objective hub-and-spoke network based on the single-distribution p-hub median problem by considering the real characteristics of CR Express in terms of cost and time. Huang et al. [45] considered the problem of multimodal transit network design in a hub-and-spoke network framework and proposed a bi-level programming model to determine the frequencies of each mode. Zhou et al. [48] studied hub-and-spoke logistics network designs considering the relationship between service pricing and the co-opetition from the perspective of duopoly logistics enterprises that set up networks jointly and allow for the transfer of both surplus capacity and carbon credits.

Nevertheless, the novel formulations for modeling multimodal hub-and-spoke network problems in real-world hazmat transport cases are not sufficient [11,49,50]. From the perspective of network design, there are still few works of literature (see, e.g., [33,51]) that simultaneously plan the hub location and route selection for hazmat multimodal transportation at the strategic level. Moreover, there is a major challenge for the risk mitigation of the hazmat transport network. Previous research [27,52] has also considered risk assessment and management but there is still room for improvement. Mohammadi et al. [53] developed a reliable multi-modal model for the transportation of hazardous material disrupted by external events and hazmat incidents. To minimize risk and cost, different transportation modes were applied and higher priorities were given to hazmat shipments. Yahyaei et al. [54] designed a backup path for each flow to form a reliable single allocation hub network under massive disruption. Jiang and Zhang [55] incorporated a parametric variational inequality that formulated the user equilibrium behavior of intermodal users in the route choice for hazmat hub-and-spoke network design decisions of the network planner to reduce the likelihood of disaster. Bianco et al. [56] considered a hazmat transportation network design problem where regional and local government authorities sought to regulate the hazmat transportation by taking into account both total risk minimization and risk equity. Brimberg [57] presented new formulations of the uncapacitated multiple allocation p-hub median problem with non-triangular constraints, which allowed for more flexible route selection with less risk. Reniers and Dullaert [58] suggested a hazmat transport security vulnerability assessment methodology based on route segments and routes to determine the relative security risk levels of the different modes of hazardous freight. Sun [32] and Abuobidalla [59] used non-linear air dispersion models, such as the box model for population and the environmental risk calculation, when planning airborne hazmat transportation. Additionally, the increased risk associated with hazmat transportation incidents [60] still calls for much work in this area, with special attention to social and environmental risks.

This study aims at enriching and enhancing the research on the problem of the hazmat multimodal hub-and-spoke network design. Additionally, the goals of this research study can be described briefly as follows: (1) the application of a hub-and-spoke network in hazmat multimodal transportation to provide a more reliable and efficient network design for the hazmat carriers at the strategic level; (2) to provide a more flexible network structure which takes detours into consideration for improving risk management performances in hazmat transportation practices; and (3) to illustrate the solution process of the problem in a clearer and more realistic way by analyzing a benchmark case, which can be more conducive for decision makers to make trade-offs between risk and cost in the planning process.

## 3. Model

### 3.1. Problem Description

For a hazmat multimodal hub-and-spoke network, there is a set of supply, demand, and candidate hub nodes. The placement decision of hubs is needed for the collection, consolidation, transfer, and distribution of hazmat from origins to destinations. Additionally, the non-hub nodes are assigned to the hubs based on the single allocation mode. Meanwhile, an optimal route for every supply to demand node via two or more hubs is determined, as shown in Figure 1. When planning the optimal route between the hubs, different transportation modes, such as road and railway, are used.

Generally, the prime objective of traditional hub-and-spoke network design for regular cargos is cost minimization. Considering that transport costs are usually positively correlated with transport distances, no more than two hubs travel in an optimal path between any pair of supply and demand nodes to ensure that the transport distance as well as cost are minimized. For example, in Figure 1, when node i is assigned to hub a and node j is assigned to hub c, the direct path a−c is better than the detour path a−b−c in terms of cost saving.

However, risk is another more important factor when designing a multimodal hub-and-spoke network for hazmat. In the real situation, risk is related to factors such as population exposure, ambient environment, and the possibility of incidents, which are not proportional to the distance. Thus, the total transport risk of a direct path between two hubs may be greater than that of a detour path. For example, the detour path a−b−c passing through a sparsely populated area may be less risky than the direct path a−c. This means that planning a route considering detours using different transportation modes may bring the advantage of risk reduction. 

In this paper, the following assumptions are made: (1) the multimodal hub-and-spoke network is a graph of G=(N, A), where $N=\left\{1,\;2,\ldots ,\;n\right\}$ represents a set of nodes while $A=\{\left(i,\;j\right):\;i,\;j\in N\}$ denotes a set of arcs. (2) The access trips of collection and distribution between non-hubs and hubs are carried out by road transportation and the transfer between hubs is either carried out by road or rail. (3) The flow from supply to demand nodes should be transported via hubs to realize the economy of scales.

### 3.2. Risk Quantification in Multimodal Transport Networks for Hazmat

Risk quantification is essential for planning and controlling the risk in a hazmat multimodal hub-and-spoke network appropriately. Additionally, the achievements of risk assessment methods for hazmat transport have been realized by many scholars [61,62]. According to the widely used definition of risk, the probability of an accident and the consequences resulting from it are two common evaluation criteria. This paper provides a unified risk evaluation method for both road and rail transport, and uses the traditional expected risk model for quantifying risk. 

The total risk in a hazmat multimodal hub-and-spoke network is divided into two parts: the transport risk and the transfer risk. 

The risk of transport on an arc can be calculated by the product of the probability of a hazmat transport accident and the consequences of the accident caused. When a hazmat accident occurs, the population in the vicinity of the accident can be seriously endangered due to factors such as explosions, wind, etc. Thus, the accident consequences are measured by population exposure in this paper. Additionally, the risk of transporting hazmat on arc (i, j) using the transport mode m, i.e., $r_{ij}^{m}$ can be calculated as follows:(1)$$r_{ij}^{m}=p_{ij}^{m}\cdot pop_{ij}^{m}$$
where $p_{ij}^{m}$ and $pop_{ij}^{m}$ are the accident probability and the number of people that may be exposed when transporting hazmat on arc (i,j) by mode m, respectively. Population exposure is calculated based on the population density and the area affected by the accident. Relative to the long distance of transportation, both trucks on the road and trains on the railway can be considered as a point on a plane [63]. Thus, the accident impact area is simplified to a rectangle and two semi-circular areas. Additionally, $pop_{ij}^{m}$ is calculated as follows:(2)$$pop_{ij}^{m}=(2d_{m}\cdot l_{ij}^{m}+\pi d_{m}^{2})\cdot \rho_{ij}^{m}$$

In Equation (2), $d_{m}^{}$ is the impact radius when using the transport mode m, $l_{ij}^{m}$ is the distance of the arc (i, j) travelled by mode m, and $\rho_{ij}^{m}$ is the density of the population in the affected area along arc (i, j) when transporting by mode m. Considering that the second part of Equation (2) actually describes the number of people affected at a hub node, which can be seen as the consequence of the transfer risk, $pop_{ij}^{m}$ is further written as Equation (3) in order to avoid calculating the consequence twice.
(3)$$pop_{ij}^{m}=(2d_{m}\cdot l_{ij}^{m})\cdot \rho_{ij}^{m}$$

Similarly, the risk of transfer is defined as Equation (4), where $p_{k}$ and $pop_{k}$ are the accident probability and the number of people that may be exposed at hub k when using transport mode m, respectively.
(4)$$r_{k}=p_{k}\cdot pop_{k}$$

Additionally, $pop_{k}$ is calculated as follows.
(5)$$pop_{k}=\pi d_{k}^{2}\cdot \rho_{k}$$
where $d_{k}^{}$ is the impact radius of an incident at node k and $\rho_{k}$ is the population density adjacent to node k.

### 3.3. Mathematical Models

Hazmat is a special class of goods with characteristics such as being flammable, explosive, corrosive, etc. In the strategic planning of a multimodal hub-and-spoke network, the risk should be reduced within a reasonable cost budget. Based on the risk quantification of the whole process of the multimodal transportation of hazmat, this paper develops a bi-objective optimization model of a single allocation hub-and-spoke network with detour for hazmat multimodal transportation which integrates the hub location as well as route and mode selection. Under the detour strategy, the flow between supply nodes to demand nodes may pass two or more hubs using different transport modes so as to reduce the total risk. Meanwhile, a counterpart model without detour is also given for comparison.

#### 3.3.1. Notations

The following notations are introduced for the model parameters and variables, as shown in Table 1 and Table 2.

#### 3.3.2. The Model of Hazmat MHSNWD 

The model of the multimodal hub-and-spoke network with detour (MHSNWD) for hazmat is constructed as follows:(6)$$\min z_{1}=\sum_{i,j,k,l\in N}\sum_{m\in M}r_{kl}^{m}\cdot f_{ij}^{}\cdot y_{ij}^{kl^{m}}+\sum_{k\in N}z_{kk}\cdot r_{k}(\sum_{i,j,l\in N}\sum_{m\in M}f_{ij}\cdot y_{ij}^{kl^{m}})$$
(7)$$\begin{array}{c} \min z_{2}=(\chi -\alpha )\sum_{i,j,k\in N}c_{ik}^{1}\cdot f_{ij}\cdot z_{ik}+(\delta -\alpha )\sum_{i,j,l\in N}c_{lj}^{1}\cdot f_{ij}\cdot z_{jl}\\ +\alpha \sum_{i,j,k,l\in N}\sum_{m\in M}c_{kl}^{m}\cdot f_{ij}\cdot y_{ij}^{kl^{m}}+\sum_{k\in K}F_{k}\cdot z_{kk}^{} \end{array}$$


*s.t.*

(8)
$$\sum_{k\in N}z_{kk}=h$$


(9)
$$\sum_{k\in N}z_{ik}=1\;\forall i\in N$$


(10)
$$z_{ik}\leq z_{kk}\;\forall i,k\in N$$


(11)
$$z_{ik}\leq y_{ij}^{ik^{1}}\;\forall i,j,k\in N:i\neq j,k\neq i$$


(12)
$$z_{jl}\leq y_{ij}^{lj^{1}}\;\forall i,j,l\in N:i\neq j,l\neq j$$


(13)
$$\sum_{m\in M}\sum_{l\in N}y_{ij}^{kl^{m}}-\sum_{m\in M}\sum_{l\in N}y_{ij}^{lk^{m}}=\left\{\begin{array}{l} 1,k=i\\ 0,k\neq i,j\;\forall i,j,k\in N,i\neq j\\ -1,k=j \end{array}\right.$$


(14)
$$\sum_{m\in M}\sum_{k\in N}y_{ij}^{kl^{m}}\leq z_{ll}\;\forall i,j,l\in N:i\neq j,l\neq j$$


(15)
$$\sum_{m\in M}\sum_{k\in N}y_{ij}^{kl^{m}}\leq 1\;\forall i,j,l\in N:i\neq j,l=j$$


(16)
$$\sum_{m\in M}\sum_{k\in N}y_{ij}^{kl^{m}}\leq z_{kk}\;\forall i,j,k\in N:i\neq j,k\neq i$$


(17)
$$\sum_{m\in M}\sum_{k\in N}y_{ij}^{kl^{m}}\leq 1\;\forall i,j,k\in N:i\neq j,k=i$$


(18)
$$y_{ij}^{ij^{m}}\leq z_{ii}+z_{jj}\;\forall i,j\in N:i\neq j$$


(19)
$$y_{ij}^{kl^{m}}\in \left\{0,1\right\}$$


(20)
$$z_{ik}\in \left\{0,1\right\}$$



Objective function (6) represents the minimization of the total risk, including the transport risk and transfer risk. Objective function (7) represents the minimization of total costs, including the transport costs and setup costs. Constraint (8) specifies that the number of hubs to be built is a pre-determined value of h. Constraint (9) ensures that each node must be assigned to only one hub node. Constraint (10) ensures that nodes in the network can be assigned to a hub only after the hub is established. Constraints (11) and (12) represent that the flow from node i to node j should be transported via arc $\left(i,k\right)$ or arc (l,j) by road when node i or node j are allocated to hub k or hub l. Constraint (13) is the flow conservation formula for the nodes. Constraints (14)–(17) guarantee that the nodes between any pair of supply and demand nodes are hub nodes. Constraint (18) prohibits the direct transport between non-hub nodes. Constraints (19) and (20) are attribute statements for binary decision variables.

In the above model, objective function (6) is non-linear because of the multiplication of two binary decision variables. To linearize the problem, a new binary variable $a_{ij}^{kl}$ is introduced and objective function (6) is changed into Equation (21) by adding a constraint, as illustrated by Equation (22).
(21)$$\min z_{1}=\sum_{i,j,k,l\in N}\sum_{m\in M}r_{kl}^{m}\cdot f_{ij}^{}\cdot y_{ij}^{kl^{m}}+\sum_{k\in N}r_{k}(\sum_{i,j,l\in N}f_{ij}\cdot a_{ij}^{kl})$$
(22)$$z_{kk}+\sum_{m\in M}y_{ij}^{kl^{m}}-1\leq a_{ij}^{kl}\;\forall i,j,k,l\in N$$

Considering that the total risk minimization is the primal objective of the design problem of a multimodal hub-and-spoke network for hazmat, the above bi-objective problem is further transformed into a single objective problem of risk minimization using the prime objective method. Additionally, the total cost is incorporated into the model as a constraint, as shown in Equation (23).
(23)$$\begin{array}{r} (\chi -\alpha )\sum_{i,j,k\in N}c_{ik}^{1}\cdot f_{ij}\cdot z_{ik}+(\delta -\alpha )\sum_{i,j,l\in N}c_{lj}^{1}\cdot f_{ij}\cdot z_{jl}\\ +\alpha \sum_{i,j,k,l\in N}\sum_{m\in M}c_{kl}^{m}\cdot f_{ij}\cdot y_{ij}^{kl^{m}}+\sum_{k\in K}F_{k}\cdot z_{kk}^{}\leq G \end{array}$$

Herein, G is the cost budget of the decision makers. Thus, the single-objective linear programming model can be solved using the optimization software CPLEX.

#### 3.3.3. The Model of Hazmat MHSNOD

In order to validate the advantages of the proposed MHSNWD model in risk mitigation, a counterpart model of the hazmat multimodal hub-and-spoke network without detour (MHSNOD) is given based on the traditional hub-and-spoke network under single allocation [14]. In the MHSNOD model, the detour strategy is prohibited, which means no more than two hubs may appear in an optimal route between any pair of supply and demand nodes. Thus, the main parameters and variables are the same, except that the decision variable $y_{ij}^{kl^{m}}$ is replaced by $v_{ijkl}^{m}$, where $v_{ijkl}^{m}=1$ indicates that the hazmat travels from node i to node j, passing node k and node l by transport mode m, and $v_{ijkl}^{m}=0$ otherwise. The MHSNOD model for hazmat is developed as follows:(24)$$\min z_{1}=\sum_{i,j,k,l\in N}\sum_{m\in M}(r_{ik}^{1}+r_{kl}^{m}+r_{lj}^{1})\cdot f_{ij}^{}\cdot v_{ijkl}^{m}+\sum_{k\in N}r_{k}\cdot \sum_{i,j\in N}f_{ij}\cdot z_{ik}$$
(25)$$\min z_{2}=\sum_{i,j,k,l\in N}\sum_{m\in M}(\chi c_{ik}^{1}+\alpha c_{kl}^{m}+\delta c_{lj}^{1})\cdot f_{ij}^{}\cdot v_{ijkl}^{m}+\sum_{k\in N}F_{k}\cdot z_{kk}$$


*s.t.*

(26)
$$\sum_{k\in N}z_{kk}=h$$


(27)
$$\sum_{k\in N}z_{ik}=1\;\forall i\in N$$


(28)
$$z_{ik}\leq z_{kk}\;\forall i,k\in N$$


(29)
$$\sum_{m\in M}v_{ijkl}^{m}\geq z_{ik}+z_{jl}-1\;\forall i,j,k,l\in N$$


(30)
$$v_{ijkl}^{m}\in \left\{0,1\right\}$$


(31)
$$z_{ik}\in \left\{0,1\right\}$$



In a similar manner, objective functions (24) and (25) minimize the total risk and total cost. Constraint (26) ensures that the number of hubs built is h. Constraint (27) ensures that each supply and demand node must be allocated to a hub node. Constraint (28) ensures that only after a hub has been created can a non-hub node be allocated to it. Constraint (26) ensures that a valid transport path with a certain transport mode exists between any pair of supply and demand nodes. Constraints (27) and (28) are attribute statements for binary decision variables.

## 4. Numerical Study

In this section, the benchmarking CAB (Civil Aeronautics Board) dataset from the OR Library is used in this study, which contains flows and costs based on the airline passenger interactions from 25 US cities. To make the solving process simpler, the original model has been simplified as a linear programming model, the solutions of which can be obtained by using the commercial software CPLEX. Meanwhile, the bi-objective problems are further transformed into single-objective problems by using the prime objective method. In the following numerical study, a detailed analysis will be developed and the solving process will be divided into two parts. For the first part, to find the solutions without a cost-budget limit, the objective of minimizing the total cost will be omitted and risk minimization will be regarded as the only objective. For the second part, the objective of minimizing the total cost will be transferred into constraints and the optimal solutions under the circumstance with cost-budget limits will be presented.

### 4.1. Parameter Settings

Referring to [40], the unit transport cost of a railway is generated by multiplying the unit transport cost of the road and the cost ratio (CR). Here, two scenarios for CR are considered: (1) CR=0.8, when the unit rail transport cost is cheaper than that of road due to its large capacity and high efficiency, and (2) CR=1.2, where the real volume of transport is small or the transport distance is short. In addition, it is assumed that the fixed set-up cost of a hub is positively correlated with the flow through the hub, i.e., $O_{i}=\sum_{i\in N}f_{ij}$. Additionally, in this study, the fixed set-up cost is generated from the interval $\left[10\log O_{i},20\log O_{i}\right]$ [41].

The population density data are from the US Federal Bureau of Statistics. Additionally, it is assumed that the density of a population near a hub is the same as that of the state in which it is located. Referring to [63], the population density in the affected area along a route is calculated by dividing the total population of the states affected by their overall area. The distances between city nodes for road and rail transportation are obtained from Google Maps and the US Federal Railroad Administration, respectively. The accident probabilities per unit distance for road and rail transportation are $p_{ij}^{1}=0.62\times 10^{-6}$ and $p_{ij}^{2}=0.19\times 10^{-6}$, respectively [30], while the accident probability of transfer is taken as $p_{k}=422.61\times 10^{-6}$ [63]. In addition, the half-width of the area affected during hazmat transportation is set as 0.8km and the impact radius during transfer is set as 2.5km [64]. The economic discount factor for collecting and distributing cargos is 1, i.e., χ=δ=1, and the economic discount factor for transfer is given as 0.2, i.e., α=0.2.

### 4.2. Numerical Results and Analysis

#### 4.2.1. Without Cost-Budget Limit

Firstly, the risk variation of the optimal MHSNWD using different numbers of hubs is explored to discuss the effect of the number of hubs on risk mitigation. To minimize the total risk, the cost budget is set to a very large value, which represents the cases with no cost-budget limit. When CR=0.8, the risk performance of the optimal network from the model of MHSNWD for hazmat is shown in Figure 2.

As seen in Figure 2a,b, the growing tendency of the transport risk is opposite to that of transfer risk when the number of hubs increases. For the transport risk, it decreases with the increase in the number of hubs. For example, when the number of hubs increases from two to eight, the transport risk decreases from 195.93 × 10^4^ to 63.54 × 10^4^ people, achieving a 68% reduction in transport risk. The transfer risk, however, increases with the number of hubs and the reduction ratio of the transfer risk is negative. As the transfer risk only accounts for a small proportion of total risk in the network, the trend of total risk is similar to that of transport risk, which also decreases with the increase in the number of hubs, as shown in Figure 2c. However, the decline rate of total risk with the number of hubs increasing tends to be slow. For example, a reduction in the total risk of up to 60% is already achieved when the number of hubs increases from 2 to 6. If the number of hubs increases to 8, only a 7% greater reduction in the total risk can be obtained.

Establishing more hubs enables the MHSNWD model to find less risky routes by providing more candidate paths. For instance, when h=2, only two hubs (8 and 20) are selected and the flow between any pair of OD nodes travels through arc (8, 20) by rail. Since there are only two hubs in the network, no detour route between hubs is available. However, when h=4, the set of hubs becomes { 1, 11, 18, 19}, which allows for more non-hub nodes to be allocated to the hubs that are located in the area of low risk. Moreover, the detour strategy permits two more hub nodes to appear in an optimal path which may bypass the densely populated areas. For example, the flow between node 12 and node 5 is transported via 12→8→20→5 when h=2 but will be diverted to a less risky route of 12→19→11→1→5 when h=4, as shown in Figure 3a,b. Apparently, the MHSNWD model for hazmat gives a more flexible option for hub placement and route planning.

Then, the risk performances of the optimal results from the MHSNWD model and MHSNOD model were compared to validate the superiority of the MHSNWD model on risk mitigation. By solving the two models, some optimal results of hub location are obtained and listed in Table 3.

As seen from Table 2, node 8 and node 20 are selected as hubs for both the MHSNWD model and MHSNOD model when h=2, which results in the same level of total risk. Since the flow between any pair of supply and demand nodes has to be realized through these two hubs, which means there exists no bypass links between them, the resulting node allocation and route selection are the same for both models.

However, in cases of setting up more than two hubs, the optimal schemes of hub location and route planning under these two strategies turn out to be different under the same number of hubs. Consequently, their performances in risk and cost management also differ, as shown in Figure 4. 

Compared to the corresponding MHSNOD, the total risk of the optimal MHSNWD is lower when h>2. Additionally, the gap between them grows significantly with the number of hubs increasing, which indicates that the detours between multiple hubs can provide great room for risk optimization. 

For example, to realize the flow between 12 and 14, the MHSNWD model and MHSNOD model give two different optimal results, as shown in Figure 5. For the optimal MHSNWD, the route 12→19→11→1→14 is selected, while in the optimal MHSNOD, the flow travels through 12→8→1→14.

Specifically, when h=8, the total risk of the optimal MHSNWD is substantially lower than that of the MHSNOD and it achieves a high reduction rate of as much as 40.67%. Additionally, it can be inferred from Table 2 that the selected Hub 25 (Washington, DC, USA) for the MHSNOD, which is a densely populated city, is a critical factor for the risk increase.

Meanwhile, it is noted that the total cost of the optimal MHSNWD with the objective of risk minimization is not always higher than that of the optimal MHSNOD with the same number of hubs. In some cases, for example, the total cost of the optimal MHSNWD is even slightly smaller than that of the optimal MHSNOD when h=3, 4, and 5, as shown in Figure 4b. This is due to the fact that the optimal MHSNWD is configured differently from the optimal MHSNOD and the optimized scheme of a design with shorter access trips brings a reduction in the transport costs of the optimal MHSNWD, which compensates for the increased costs associated with inter-hub detour to some extent.

#### 4.2.2. With Cost-Budget Limit

Risk minimization is a key objective in the design of a hazmat multimodal network; however, hazmat carriers should also take cost budgets into consideration in order to gain a competitive advantage. As we know, although an increase in the number of hubs can achieve risk reductions, it can also lead to a high fixed set-up cost at the same time. In addition, the selected route by the MHSNWD model may lead to a higher transportation cost for certain scenarios. Therefore, it is necessary to investigate the effect of cost budget on the risk performance of the network to inform decision makers on how to balance risk and cost.

In this study, the risk performances of the optimal MHSNWDs with cost-budget constraints were compared with that of the counterpart optimal MHSNODs under the same condition. We first solved the optimization problem of the multimodal hub-and-spoke network design with the objective of cost minimization and found the corresponding total cost. Then, by taking it as the cost-budget constraint, we resolved the problem to minimize the total risk. By increasing the cost budget at a rate of 5% within the acceptable range of the decision makers, we repeated this procedure to obtain the total risk value of the optimal MHSNWD and MHSNOD. Furthermore, two scenarios using CR=0.8 and CR=1.2 were studied to find some managerial insights on developing multimodal transportation.

Assuming that the number of hubs to be established is four, the total risk variations of the optimal MHSNOD and MHSNWD network with a cost budget under different cost ratios are given in Figure 6.

As seen from Figure 6, the optimal schemas for the cost minimization of the MHSNWD and MHSNOD are the same, resulting in an identical performance of total risk, whereas their total risks all decrease when the cost budget expands and levels off at the minimum risk value in both scenarios. Therefore, decision makers can choose the highest acceptable cost budget to take less risk according to their preferences.

In addition, the MHSNWD model can achieve a consistently lower value of total risk than that of the MHSNOD model when increasing the cost budget by the same percentage. For example, in Figure 6a, when investing 5% more in the cost budget, the total risk of the optimal MHSNWD decreases from 186.14 to 128.66, which is a reduction of almost 30.88%. Comparatively, the optimal MHSNOD only achieves a risk reduction of 23.98% from 186.14 to 141.49. If the increase percentage reaches 10%, the optimal MHSNWD and MHSNOD will reduce the total risk by 39.12% and 25.29%, respectively. This also implies that the MHSNWD exceeds the MHSNOD in risk control.

Meanwhile, it has been found that the transportation mode associated with a certain unit of transport cost and accident probability has a great effect on the network design and final risk performance. When CR=0.8, the minimum total cost of the optimal network is 934.99 and the corresponding total risk is 186.14. In the case of CR=1.2, their values reach 936.73 and 308.58. Apparently, the rise in railway cost not only increases the total cost of the optimal network but also causes the reconfiguration of the network and transportation mode, resulting in higher total risk.

In comparing Figure 6a with Figure 6b, it can be seen that the total risk of the optimal network under CR=1.2 is always higher than that under CR=0.8 when the cost budget is increased by the same percentage for both the MHSNOD and MHSNWD models. In addition, in the case of CR=0.8, the lowest network risk is achieved when the cost budget is expanded by 15% from the optimal cost, while in the case of CR=1.2, the cost budget needs to be expanded by 25% from the optimal cost to minimize the network risk value. This means that the decision maker needs to invest more in the cost budget in order to reduce the risk to the same level as in the case of CR=1.2.

Figure 7 and Figure 8 show the network optimization results for the MHSNOD model and MHSNWD model when the cost budget is expanded by 5% and 10%, respectively, in the cases of CR=0.8 and CR=1.2.

Under certain CR, the results of the MHSNWD model and MHSNOD model both differ in hub locations under the same cost budgets and increase at a steady rate of 5%. More importantly, the links between hubs for different networks vary significantly. In the MHSNOD model, the network is modelled in such a way that there must be a direct link between each pair of hubs, as shown in Figure 7a,c. Additionally, in this case, the links between all four hubs are by rail. In contrast, for the MHSNWD model, which has a more flexible transportation strategy, two hubs will not be directly connected if the risk of a direct path is higher than a detour path. For example, in Figure 7b, Hub 12 is not directly connected to Hub 18 but arrives at Hub 18, passing by two more hubs, namely Hub 11 and Hub 5. And in both cases as shown in Figure 7b,d, it is favorable to build more hubs in less densely populated areas when decision makers are willing to invest more to reduce the total risk of the network.

In addition, a comparison between Figure 7 and Figure 8 shows that the connections between the hubs occur by rail when CR=0.8, while in the case of CR=1.2, some of the hubs are connected by road. This is due to the fact that with rail transport costs being higher than road transport costs, cutting down the cost budget leaves some of the hubs to be connected by the cheaper road transport method. In particular, when the cost budget is increased by only 5% of the minimal cost, a great quantity of the routes are connected by road. However, when the cost budget is increased by 10%, more routes are connected by railway, which may lead to lower risks. For example, in Figure 8b, all hub connections are realized by road, except for the link between Hub 9 and Hub 12. However, in Figure 8d, the hubs are all connected by railway. Therefore, the increase in cost budget allows for the selection of rail transportation with a lower accident probability, thus reducing the total network risk.

## 5. Discussion and Managerial Implications

First, this study contributes to the literature on the application of a hub-and-spoke network in the multimodal transportation of hazardous material. Although the point-to-point direct transport network is often taken for granted in the prevailing literature, it has been challenged by the higher efficiency and reliability of the hub-and-spoke network. Therefore, it is meaningful to apply this network in the long-distance multimodal transportation of hazmat [6,11] and the combinatory application can also provide helpful suggestions, for instance, regarding integrating the hub location and route selection to better control the total risk at the strategic level.

Second, this study considers the detour strategy for improving risk management performances in hazmat transportation practices. It breaks the common assumption that there should be no more than two hubs in the same transportation process, which can greatly shorten the transportation distance while not being conducive to risk mitigation. Based on the detour strategy, the developed MHSNWD model allows for more than two hub nodes to be selected to form an optimal path between supply and demand nodes, which significantly reduces the transport risk by inter-hub detours. Although the increase in the number of hubs may lead to a slight increase in transfer risk, the total risk of the MHSNWD is markedly lower than that of the MHSNOD. Therefore, the proposed MHSNWD model is more flexible and reasonable, and the results will help carriers to improve risk performances on a managerial level. 

Third, this paper implements the proposed model on a benchmark case and presents solutions numerically. Since simple model solving may not provide enough reference for practical application, a numerical study has been analyzed and discussed in a previous section. By comparing the solutions under different situations, the results showed some useful insights regarding deploying the hazmat multimodal network and provided more valuable implications for carriers in terms of both risk mitigation and the trade-off between risk and cost. 

To summarize, we have found the managerial implications to be as follows: (1) to better manage the risks in the multimodal transportation of dangerous goods, carriers could consider the application of a hub-and-spoke network with detour when creating the transport strategy; (2) a reasonable increase in the number of hubs is meaningful because it can provide more choices for the allocation decision and route selection between supply and demand nodes and hubs, which may avoid passing through areas with high risk; and (3) further reductions in transportation costs, such as railway costs, can increase the use of rail transport with a low accident probability. Therefore, decision makers should scientifically develop and plan various modes of transportation and set the cost budget properly according to their risk preferences.

## 6. Conclusions

Hazmat accidents pose a significant threat to the public, causing economic losses, casualties, environmental pollution, and other social problems. Additionally, there is a wide awareness among stakeholders, such as hazmat carriers and the government, that a reliable and efficient multimodal hub network for hazmat is of great significance to promote the economic development of its service areas. In this paper, we explored the problem of designing an optimal multimodal hub-and-spoke network with detour for hazmat, filling the gap in the hub-and-spoke network design of hazmat. From the perspective of strategic network planning, it integrates the hub location and route selection of a MHSNWD to reduce the total risk. Furthermore, a numerical study based on the CAB dataset was given to obtain some managerial insights. However, limitations still exist in our research, which leaves a wide scope for future study.

The authors applied the model to a benchmark case and not to real data. Therefore, it is uncertain whether the solution generated is also valuable in real life applications. If real applications can be studied in future work, the solution could be verified and the study will be more complete on a theoretical level. Moreover, the method of risk assessment in this paper lacks consideration of dynamic factors such as market changes and environmental changes. According to Mahmoudabadi et al. [65], future studies could consider the dynamic factors and redefine the calculation method of risk value. Furthermore, more efficient algorithms can be designed to improve the problem-solving efficiency.

## Figures and Tables

**Figure 1 ijerph-18-12470-f001:**
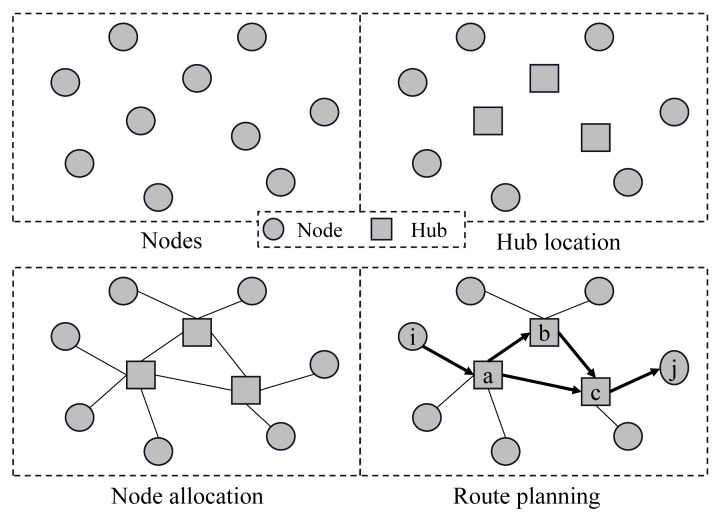
Schematic diagram of a hazmat multimodal hub-and-spoke network with detour.

**Figure 2 ijerph-18-12470-f002:**
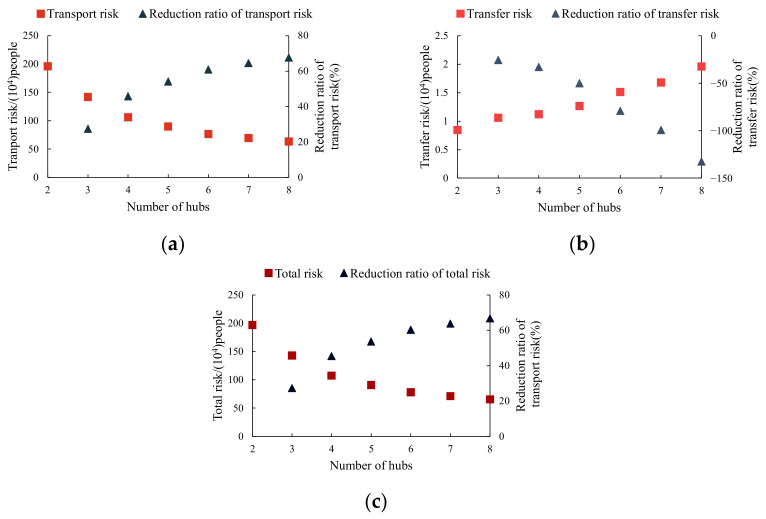
The risk performance of the optimal MHSNWD: (**a**) transport risk, (**b**) transfer risk, and (**c**) total risk.

**Figure 3 ijerph-18-12470-f003:**
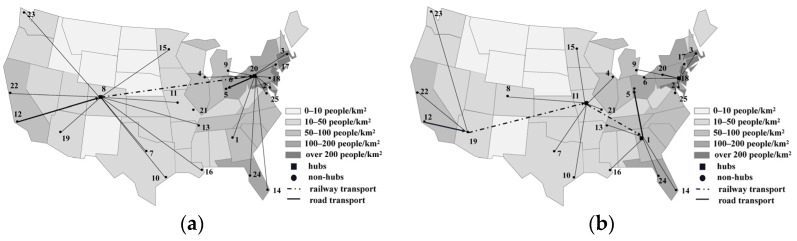
The optimization result of MHSNWD for hazmat when (**a**) *h* = 2; (**b**) *h* = 4.

**Figure 4 ijerph-18-12470-f004:**
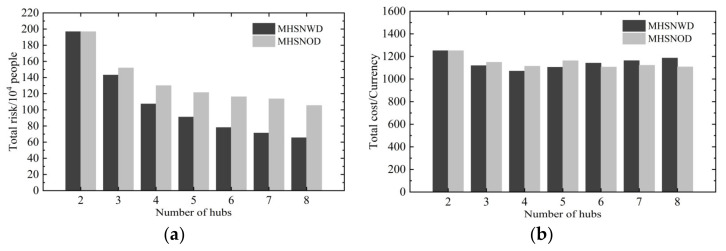
The performance of the total risk and cost for MHSNWD and MHSNOD network: (**a**) total risk and (**b**) total cost.

**Figure 5 ijerph-18-12470-f005:**
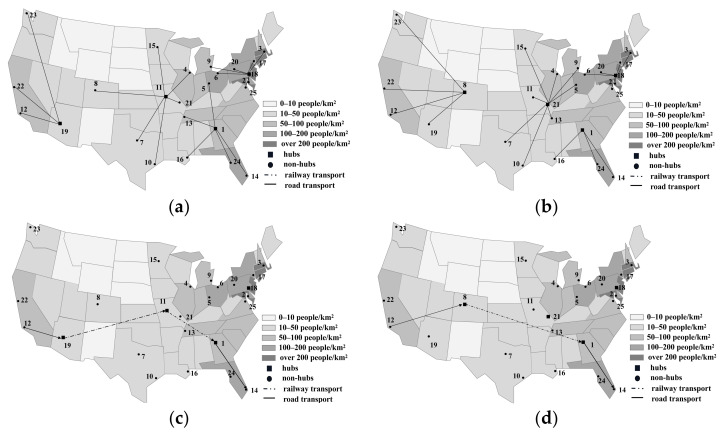
Comparison of the optimal IHSNWD and IHSNOD network when h=4. (**a**) Location and allocation of the MHSNWD. (**b**) Location and allocation of the MHSNOD. (**c**) Routing of the MHSNWD. (**d**) Routing of the MHSNOD.

**Figure 6 ijerph-18-12470-f006:**
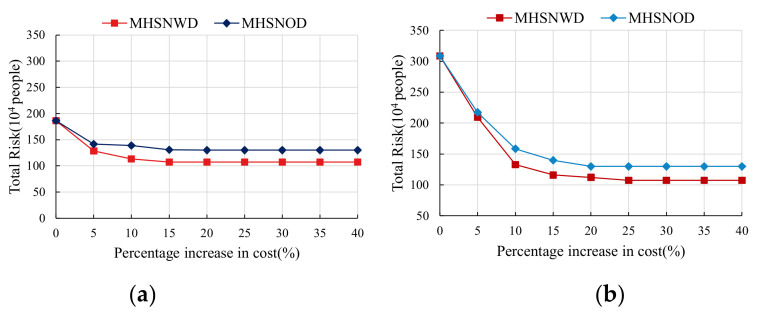
Comparison of the total risk variations of MHSNOD and MHSNWD under different *CR*s: (**a**) CR = 0.8 and (**b**) CR = 1.2.

**Figure 7 ijerph-18-12470-f007:**
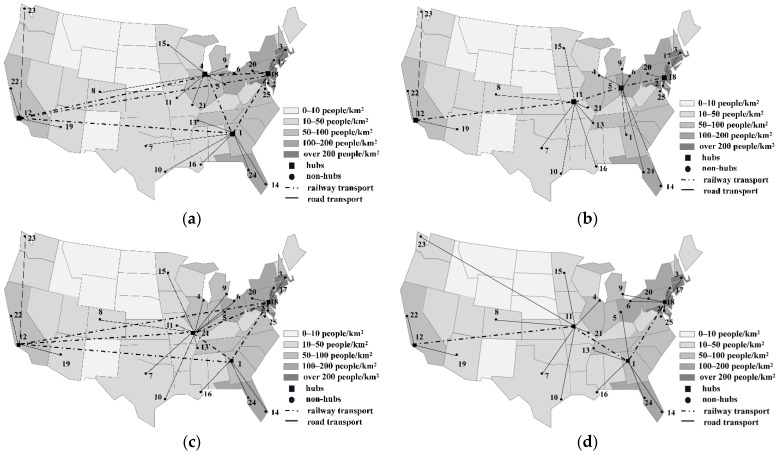
The optimization results when CR=0.8: (**a**) MHSNOD (5% increase in cost); (**b**) MHSNWD (5% increase in cost); (**c**) MHSNOD (10% increase in cost); and (**d**) MHSNWD (10% increase in cost).

**Figure 8 ijerph-18-12470-f008:**
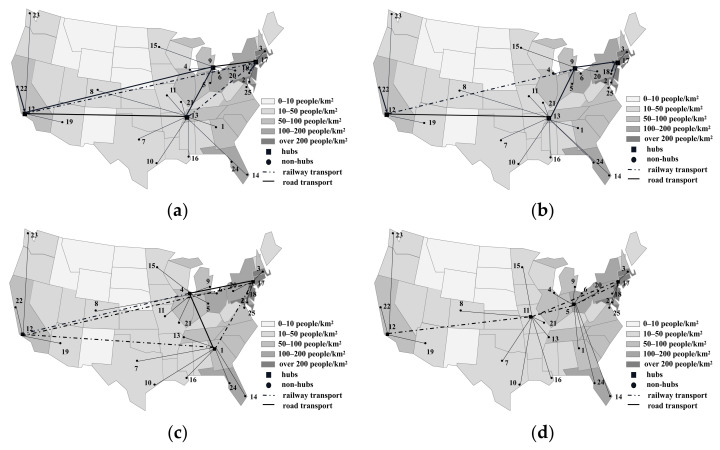
The optimization results when CR=1.2. (**a**) MHSNOD (5% increase in cost), (**b**) MHSNWD (5% increase in cost), (**c**) MHSNOD (10% increase in cost), (**d**) MHSNWD (10% increase in cost).

**Table 1 ijerph-18-12470-t001:** Parameter definitions for the model.

Parameters	Definitions
G=(N, A)	A complete graph for the hazmat multimodal hub-and-spoke network
$N\;=\;\left\{1,\;2,\;\ldots ,\;n\right\}$	A set of supply, demand, and candidate hub nodes
$A=\{\left(i,\;j\right):\;i,\;j\in N\}$	A set of arcs or paths
$M\;=\;\left\{1,\;2\right\}$	A set of transport modes, where 1 represents by road and 2 represents by rail
m	Modes of transport
$f_{ij}$	Flow between node i to node j
χ, α, δ	Discount factor for the collection, transfer, and distribution of hazmat, where α<χ and α<δ
$F_{k}$	Setup cost of hub k(k∈N)
$r_{k}$	Unit risk of the transfer of hazmat through hub k
$c_{ij}^{m}$	Unit transport cost by mode m from node i to node j
$r_{ij}^{m}$	Unit transport risk by mode m from node i to node j
h	Number of hubs

**Table 2 ijerph-18-12470-t002:** Decision variables for the model.

Variables	Definitions
$z_{ik}$	$z_{ik}=1$ indicates that node i is assigned to hub k $,\;\mathrm{otherwise}\;z_{ik}=0$
$y_{ij}^{kl^{m}}$	$y_{ij}^{kl^{m}}=1$ indicates that the hazmat travels from node i to node j by transport mode m through the arc (k, l) $,\;\mathrm{otherwise}\;y_{ij}^{kl^{m}}=0$

**Table 3 ijerph-18-12470-t003:** The optimal hub location results from MHSNWD and MHSNOD under different h.

h	MHSNWD Model	MHSNOD Model
2	8, 20	8, 20
3	1, 18, 19	1, 8, 18
4	1, 11, 18, 19	1, 8, 18, 21
5	1, 11, 18, 19, 20	1, 8, 18, 20, 21
6	1, 4, 11, 18, 19, 20	1, 4, 11, 12, 17, 20
7	1, 4, 11, 12, 18, 19, 20	1, 4, 8, 11, 12, 17, 20
8	4, 11, 12, 14, 16, 18, 19, 20	1, 4, 6, 8, 11, 12, 17, 25

## Data Availability

The data presented in this study are available upon request from the authors.

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
