# Peer review of "Design Optimization of a HAZMAT Multimodal Hub-and-Spoke Network with Detour"

_ijerph, 2021, doi:10.3390/ijerph182312470_

Round 1
Reviewer 1 Report
Report on Design Optimization of HAZMAT Multimodal Hub-and-spoke Network with Detour (ID: IJERPH-1467199) for International Journal of Environmental Research and Public Health
This paper considers a transport network to find the “best” route to deliver hazardous goods. The authors overall have done a good job. I find the paper well-presented and I do think that the paper should be published.
The model specification is clear. The authors have formulated two different possible frameworks of multiple Hub and Spoke networks, namely, MHSNWD (with detour) and MHSNOD (without detour). The constrained optimisation problems in Pages 7 and 8 are well-developed.
The authors have solved a real case to present their solutions numerically. The analysis and the results therefore are well taken up, although with hardly any real surprises at all!!
My only reservation is that there is no attempt of solving these optimisations in an analytical way; It would have been useful to have at least some discussion on actual process of solving such a problem. Numerical solutions and simulations are nice to understand but we do need some general mathematical result for any such problems. I would suggest that the author at least offer something in this respect.
I did follow all the notations, however, I have a reservation for using “p” as the number of hubs! Perhaps a different notation (such as h) would have been better?!
An editorial check would always be healthy; although, I must say that I didn’t find any typos in this version. Still, one must carefully check and eradicate the odd grammatical mistakes present, along with punctuations and particularly, spacing between lines and words in many places (including the reference list).
To sum up, I do think this paper has discussed something nice and thus should be accepted for publication, however, should present some general thoughts on the optimisation process.
Reviewer 2 Report
This manuscript is overall structured very well and is easy to read, however some improvements are needed before publication.
The authors claim that research gaps are provided in section 2, among other things, however, it is necessary to highlight these gaps very well by referring to the most recent literature.
In the literature review, the authors refer to sustainable development, so there is a need to supplement the literature review with references to the topic of sustainability in risk management:
https://doi.org/10.3390/su13063277
https://doi.org/10.1007/s40171-021-00277-7
https://doi.org/10.1016/j.jclepro.2020.121579
https://doi.org/10.3390/su13158196
Deepening also the aspect of risk assessment:
https://doi.org/10.1002/bse.2520
https://doi.org/10.1007/s10098-020-01901-3
https://doi.org/10.1016/j.jhazmat.2020.123186
Finally, in the conclusion the authors should stress the limitations of their study, the managerial implications in addition to the theoretical implications that should fill the gaps that arose in the literature review.
Reviewer 3 Report
In the article, the authors presented a novel optimization model of multimodal hub-and-spoke network with detour (MHSNWD) for hazmat on the strategic level with is designed as a reference for decision-making planning in the hazmat transportation network.
Next you may find some comments (strengths, weaknesses and recommendations).
Strenghts:
- Title of the paper is clear.
- The authors use a large literature in the field and the references are adequate to the topic.
- The method used is correct.
Weaknesses:
- No clearly articulated goal of the work
- There is no discussion section
In addition, the authors presented a model that helps to reduce the total risk of the whole process of multimodal transport of hazmat. As the authors emphasize: "Hazmat accidents pose a significant threat to the public, causing economic losses, casualties, environmental pollution and other social problems. And there is a wide awareness among stakeholders such as hazmat carriers and the government that the reliable and efficient multimodal hub network for hazmat is of great significance to promote the economic development of its service areas".
The paper is good and I believe that it represents a substantial contribution to the literature and is within the scope of the journal of international repute such as " International Journal of Environmental Research and Public Health" and I recommend this paper for publication after addressing the following minor issues.
Recommendations:
I miss some information relative to the authors contribution to the literature. Please, mention what you bring new with your research.
Regarding the Conclusion section. It is recommended to carry out research on the verification of the developed model in practice. Can you mention if this is a limit of your research, or you intend to develop the research in a future study?
Round 2
Reviewer 2 Report
Dear Authors,
I have read your revision appreciating the improvement effort you have made. I therefore consider this latest version of the manuscript suitable for publication.
Congratulations!